# A Systematic Review of the Dietary Choline Impact on Cognition from a Psychobiological Approach: Insights from Animal Studies

**DOI:** 10.3390/nu13061966

**Published:** 2021-06-08

**Authors:** Fernando Gámiz, Milagros Gallo

**Affiliations:** Department of Psychobiology, Institute of Neurosciences, Center for Biomedical Research (CIBM), University of Granada, 18010 Granada, Spain; mgallo@ugr.es

**Keywords:** anxiety, attention, behavior, choline supplementation, choline deprivation, emotion, learning, memory, rodent

## Abstract

The influence of dietary choline availability on cognition is currently being suggested by animal and human studies which have focused mainly on the early developmental stages. The aim of this review is to systematically search through the available rodent (rats and mice) research published during the last two decades that has assessed the effect of dietary choline interventions on cognition and related attentional and emotional processes for the entire life span. The review has been conducted according to Preferred Reporting Items for Systematic Reviews and Meta-Analyses (PRISMA) statement guidelines covering peer-reviewed studies included in PubMed and Scopus databases. After excluding duplicates and applying the inclusion/exclusion criteria we have reviewed a total of 44 articles published in 25 journals with the contribution of 146 authors. The results are analyzed based on the timing and duration of the dietary intervention and the behavioral tests applied, amongst other variables. Overall, the available results provide compelling support for the relevance of dietary choline in cognition. The beneficial effects of choline supplementation is more evident in recognition rather than in spatial memory tasks when assessing nonpathological samples whilst these effects extend to other relational memory tasks in neuropathological models. However, the limited number of studies that have evaluated other cognitive functions suggest a wider range of potential effects. More research is needed to draw conclusions about the critical variables and the nature of the impact on specific cognitive processes. The results are discussed on the terms of the theoretical framework underlying the relationship between the brain systems and cognition.

## 1. Introduction

The interest for improving cognitive functions by dietary supplementation has increased over the last decades. However, the evaluation of human studies is hindered by the difficulty of controlling a number of social, educational and economic factors. It is also difficult because a wide variety of both supplemented diets with various combinations of ingredients and behavioral indexes are used [1]. Even though animal studies can overcome some of these difficulties regarding the selection of the specific dietary intervention and the control of confounding variables, they often lack a systematic approach based on a deep knowledge of the cognitive functional architecture. In fact, cognition is a poorly defined term which includes a variety of functions such as attention, learning and memory. Moreover, each of these functions can also be dissociated in components depending on particular brain networks with different developmental courses. It has been pointed out that this makes it difficult to draw conclusions based on comparisons between different studies [2], and it is necessary to define the processes assessed by a particular behavioral test taking into account the underlying theoretical framework [3]. 

In animal studies, cognition refers to the performance in learning tasks that, in addition to memory, also require the contribution of several processes such as attention, motivation, arousal and emotion. With respect to memory, independent mechanisms have been proposed for working memory, which consists of the ability to temporally hold and manipulate a limited amount of information. In rodents, working memory is assessed in tasks relying on the information available within a testing session but not between sessions [4], and this has been related with the prefrontal cortex function [5], although other brain areas cannot be excluded [6]. Regarding long-term memory, a dissociation between hippocampal-dependent declarative/relational memory versus nondeclarative forms of memory which depend on a variety of different brain systems is widely accepted ([7], for a review). In rodents, spatial learning tasks are frequently used to assess declarative memory. The ability for spatial navigation involving the formation of complex spatial representations is based on the establishment of relationships amongst several external cues. This spatial strategy to solve a navigation task requires hippocampal integrity, but other nonhippocampal-dependent strategies based on simple associative learning or habit formation may also be used if individual cues signal the target location. Context learning and other learning tasks including contextual information also involve relational memory requiring the hippocampal involvement. Likewise, the ability to remember a previously exposed item is assessed in rodents in the object recognition memory task. This task requires the integrity of brain regions adjacent to the hippocampus, such as the perirhinal cortex or even the hippocampus, if the procedure involves contextual or temporal information [8]. Moreover, a brain region might contribute to various forms of learning. In fact, although the amygdala is critical for fear conditioning in rodents [9], it also plays a role in declarative and nondeclarative memory, provided that emotionally arousing learning tasks are used. Therefore, the evaluation of the diet–cognition relationship in animal research needs to take into account the analysis of the learning and memory tasks used.

Choline, which is among the micronutrients included in supplemented diets, has been previously related with cognition. Choline is an essential nutrient requiring appropriate dietary levels to complement the endogenous synthesis in the liver and brain [10]. Dietary choline availability has an impact on general health and specifically on the brain functions involved in cognition. It plays a role in maintaining cell membranes and myelination throughout the synthesis of phospholipids, as the precursor of acetylcholine affects cholinergic neurotransmission and it contributes to epigenetic changes by the methyl metabolism through betaine and the homocysteine reduction [11,12]. In particular, the basal forebrain cholinergic system plays a relevant role in attentional modulation of the frontoparietal cortex [13] as well as in the hippocampal memory functions [14,15]. In addition, cholinergic neurotransmission in the amygdala is involved in anxiety-related tasks [16]. Figure 1 shows the main functions of dietary choline in the nervous system. In addition to its role in the maintenance of cell membranes and epigenetic processes, its influence on brain acetylcholine levels has been associated with different cognitive processes mediated by several brain areas (amygdala, hippocampus, prefrontal cortex, etc.) and brain circuits.

Previous reviews have focused on the different effects of the availability of choline in cognition and neurological health both in humans and rodents at different stages of the life span. However, there has been particular interest in early-developmental studies as demands for choline increase during gestation and lactation [10,11,17,18,19]. In fact, the most comprehensive review of the research carried out on animals was published fifteen years ago, and it focused on early-development interventions [10]. The evidence reviewed supports a possible causal relationship between choline supplementations during development, enhanced cognitive performance and changes of the brain function that pointed mainly to the hippocampus and the cholinergic system. The cognitive enhancement was especially evident in complex demanding tasks interpreted by the authors as more difficult and at advanced ages. The authors also noted the absence of studies establishing dose-response curves, since all of the studies reviewed used a single dose of choline. A more recent systematic review also included human studies and surveys of the last 10 years of animal research, focusing on the effects of prenatal and postnatal maternal choline supplementation [11]. Both animal and human research indicated the relevance of maternal and child choline supplementation during the first 1000 days of life for supporting brain and cognitive development, as well as protection from neural insults such as alcohol exposure or neurodegeneration. Regarding cognitive assessment, the eight animal studies reviewed reported beneficial effects of choline supplementation in memory tests such as the object recognition memory tasks. However, although some human randomized well-controlled studies have reported improved color-location memory and processing speed, most of the trials and observational studies found no relationship between the maternal choline intake and offspring intelligence or cognition. Contrary to animal research, choline supplementation doses varied amongst the human studies, pointing to long-term beneficial effects of high choline dosages. Therefore, the authors recommended revision of the recommendations regarding the adequate choline intake during development, and they highlighted the need for dose-response studies. Blusztajn et al. [20] also reviewed human and rodent research with an emphasis on the neuroprotective effects of dietary choline in pathological conditions such as epilepsy, schizophrenia, Alzheimer’s disease and Down and Rett syndromes. Both the animal and the human studies pointed to the beneficial effect of choline supplementation during gestation and early postnatal period on cognition related with the development of the hippocampal function. The authors also reviewed emerging evidence from human studies to propose a potential benefit of adult choline intake mediated by the phospholipid metabolism for preventing the memory impairment induced by aging and Alzheimer’s disease. Other systematic reviews relating dietary choline and several health outcomes which include cognition have surveyed human observational and intervention studies [21,22]. 

However, systematic reviews of the rodent studies relating the availability of choline throughout the whole life span and cognition are lacking. Given that the attempts to establish a potential relationship between choline availability and cognition in adults do not seem to be conclusive, it seems valuable to update the results obtained in animal studies throughout the life span. Moreover, the available reviews centered on early development have not categorized the behavioral tests applied according to the specific cognitive processes involved, although there is a tendency to interpret the cognitive improvement associated to choline supply with changes of the hippocampal system function. However, given the multiple mechanisms through which dietary choline can affect the brain function (Figure 1), a more general effect extended to nonhippocampal-dependent cognitive processes can be hypothesized. A revision of the behavioral outcome based on an updated classification of the cognitive processes involved would help to draw a wider and more accurate picture. Thus, the aim of this review is to systematically search for the available rodent research reports published during the last two decades that have assessed the effect of dietary choline interventions on cognition throughout the developmental stages, including prenatal, perinatal, adolescence, adulthood and aging, with special emphasis on analyzing the behavioral tasks applied.

## 2. Materials and Methods 

### 2.1. Design

This systematic review was conducted according to Preferred Reporting Items for Systematic Reviews and Meta-Analyses (PRISMA) statement guidelines [23]. We have included peer-reviewed research articles that explore the effect of dietary choline interventions (through supplemental or deficient diet) on cognition. Our analysis has focused on rodent models (rats and mice) and dietary choline intervention. Given the date of previous reviews on related issues the search covered the last two decades. Therefore, a systematic literature search was conducted in April 2020 and updated in March 2021 using the databases Pubmed (All databases) and Scopus. 

### 2.2. Inclusion and Exclusion Criteria

The inclusion criteria to select the studies included in this systematic review were as follows:Publication date 2000–2020, inclusive.Experimental studies applying dietary choline intervention.Animal studies using rats or mice as subjects.Assessment using, at least, one cognitive-behavioral task.Publication in English.The exclusion criteria were as follows:
(1)Reviews, systematic reviews or meta-analysis(2)Insufficient or deficient information on the methodology or statistics applied.(3)The administration of dietary choline in combination with other ingredients that might confuse the effect of the choline intervention.(4)Cognitive-behavioral data already published in a previously selected article to avoid duplication.

### 2.3. Search Strategy and Data Extraction 

Searching covered experimental studies included in PubMed and Scopus from January 2000 up to December 2020. The key terms applied included: (choline or cholin*), (diet or dietary), (cognition or learning or memory) and (rat or mice or mouse) in the title, abstract or keys words. 

Below are the specific terms used in our bibliographic search: 

Pubmed: (choline or cholin*) and (diet or dietary) and (cognition or learning or memory) and (rat or mice or mouse) Filters: English, from 2000–2020.

Scopus: (TITLE-ABS-KEY (cholin*) AND TITLE-ABS-KEY (diet OR dietary) AND TITLE-ABS-KEY (cognition OR learning OR memory) AND TITLE-ABS-KEY (rat OR mice OR mouse)) AND PUBYEAR = 2000–2020 AND (LIMIT-TO (DOCTYPE, “ar”)) AND (LIMIT-TO (LANGUAGE, “English”)).

After excluding duplicates, the title and abstract were screened, and the inclusion/exclusion criteria applied. The selected articles were read, and those meeting the criteria were used for data analysis. Data extraction was performed by two independent researches based on a coding template that included the following items: number, type, sex and age of the experimental subject; choline doses and administration procedure; treatment duration and age of administration (prenatal or postnatal); cognitive process assessed and task used, previous pathological condition of the animal models used, effect of dietary choline supplementation; effect of dietary choline deficiency and no effect of dietary choline intervention. The detailed PRISMA flowchart showing the selection procedure is presented in Figure 2 [24]. In brief, we obtained a total of 651 articles from both databases. After the exclusion of duplicates, the initial database search yielded 363 potentially relevant citations, of which 65 were retained for full-text review. Finally, 44 articles matched the inclusion criteria.

In addition, in order to perform the general description and bibliometric analysis of the studies found, the *Vosviewer* software was used. This is a useful tool for constructing and visualizing bibliometric networks developed by Nees Jan van Eck and Ludo Waltman from Leiden University [25].

## 3. Results

### 3.1. General Description of Data

Table 1 summarizes the results of the 44 articles included in the systematic review based on our coding template. The effect of dietary supplements including choline on cognition is a research topic that has received increasing interest in recent years amongst researchers using rodent models, as can be seen in Figure 3, which represents the number of articles published per year according to the first search performed. However, after filtering based on the inclusion/exclusion criteria, the number of articles published decreased and remained stable over the years at an average of 2.09 papers per year. Given that our selection criteria are more restrictive than the general topic, we find it interesting to note how the effect of dietary choline on cognitive performance in rodent models maintains a constant publication rate while there is an increase in research on the general topic. The fact that the number of articles centered on the specific impact of choline that met rigorous criteria had not increased in the last two decades points to the need to review the current knowledge in order to draw well-founded conclusions.

Regarding the bibliographic characteristics of the 44 articles that compose this systematic review, they have been published in 25 different scientific journals. Amongst them, Brain Research with 7 papers published can be highlighted, followed by Nutritional Neuroscience (4 papers) and Behavioral Brain Research (4 papers). Moreover, the Brain Research papers received a total of 182 citations, followed by Neurotoxicology and Teratology (123 citations in 2 articles) and Birth Defects Research part A: Clinical and Molecular Teratology (106 citations in only 1 article published) (Figure 4).

A total of 146 authors have been involved in the publication of these 44 articles. Jan Krzystof Blusztajn, Christina L. Williams, Isabel de Brugada and Warren H. Meck are the authors with the most articles published on the topic (7 each). The analysis of the number of publications per author and their co-authorship shows three large groups, represented by JK. Blusztajn, I. de Brugada and Ramón Velazquez (Figure 5).

Taking into account the total number of citations, the three most cited authors are JK. Blusztain (332 citations, average of 47.4 per paper), CL. Williams (258 citations, average of 36.8 per paper) and Jennifer D. Thomas (229 citations, average of 76.3). Moreover, of the 44 articles included in our database the three most cited articles are Thomas et al. 2010 [52] (106 citations), followed by Thomas et al. 2000 [68] (104 citations) and Mellot et al. 2004 [64] (79 citations) (Figure 6). A total of 1123 authors have been cited in the 44 papers selected; the three with the most citations are WH Meck (146), Steven H. Zeisel (57) and JD Thomas (40).

### 3.2. Animals: Strain, Sex and Age

The majority of the studies (35 of the 44) used rats, mainly Sprague-Dawley (*n* = 23) and Wistar (*n* = 9). Mice models were used exclusively to explore the effect of supplementation on pathologies. All the mice studies except one [30] were transgenic models of different conditions such as APP/PS1 mice for Alzheimer’s disease [26,27,28], Ts65Dn mice for Down’s syndrome [35,39,42,51,70] and BTBR T + Itpr3tf/J mice for autism [37]. Gitik et al. [30] assessed the effect of choline supplementation on C57BL/6J mice exposed to nicotine during adolescence.

Regarding sex, 8 of the 44 studies used both males and females [26,32,37,52,54,56,59,64,65,68], whilst in only two studies, all the subjects were females [27,46]. The rest of the studies (*n* = 33) included only males, and only one did not specify the sex [36]. 

The age at which the animals were tested varied amongst the studies. This factor is of relevance when interpreting the results. Thus, although most of the results were obtained in adult animals, the age at testing varies from 2 to 24 months. The majority of these studies used young-adult 2–7 month old subjects, but subjects older than 8 months could be considered mature-adult [26,27,28,35,39,41,42,51,54] and those older than 20 months as aged [56,59]. A few studies investigated the effect of prenatal interventions on the cognitive abilities of preweaning or early postweaned [29,37,64] and adolescent subjects about 28–42 days of age [36,56,58,63,69] according to the strict adolescence criteria based on behavioral discontinuities [71]. Similarly, Wainwright et al. [72] explored the effect of early postnatal choline supplementation from postnatal days (PND) 5–18 in artificially reared rats when the animals reached adolescence. We identified only one study that explored the effect of choline supplementation during adolescence [58]. The authors maintained adolescent 28 day old rats on a choline supplemented diet for a period of 14 days before they were subjected to brain injury and behaviorally tested 8 days after.

### 3.3. Choline Supply: Administration Procedure and Dose

The great majority of the revised studies investigated the effect of choline supplementation supplied either in dry diet or diluted in water. Eight of them also included a choline-deficient group for comparison. We identified only three studies aimed at investigating the effect of choline deficiency so that they only included groups receiving diets containing 0 g/kg [50,54,67]. This can be attributed to the already well-known deleterious effect of choline deficiency on cognition, thus reducing the interest of the researchers. 

The majority of the studies found delivered choline supplementation in palatable dry diets. Most of them applied choline chloride supplemented diets containing about 4.5 times the choline content of the standard diets. All of them, except [30] who used a supplementation of 9 g/kg versus 2 g/kg in the standard diet, compared 5g/kg versus 1.1 g/kg. A few studies used diets with higher choline chloride doses containing seven times [47,49] or ten times [58,62] as much choline as the standard diet. We identified only one study [61] that applied a lower supplemented diet (0.5–1 g/kg). 

Eight studies delivered the choline supplementation in the drinking water, often sweetened in order to mask the bad taste. Most of them used a 25 mM concentration equivalent to 3.5 g/kg. We identified only two studies using choline intubation [52,68]. 

Therefore, the fact that most of the studies that used the same administration procedure applied similar doses favors comparisons between the results obtained. However, none of the studies assessed the changes induced by the treatment in the plasma choline levels. Only those studies with the intention of exploring the brain changes associated with the cognitive effects of choline supplementation reported higher cerebral choline levels [36], increased number of cholinergic neurons [28,35] and changes in cholinergic neurotransmission related with the cholinergic nicotinic receptors [58,62].

### 3.4. Choline Intervention: Timing and Duration

Most of the studies applied choline supplementation or deprivation early during development, either prenatally or during infancy and adolescence. With regard to the prenatal interventions, several studies covered either particular gestational periods or the entire gestation period [36,51], and nine studies even continued the prenatal treatment throughout the postnatal period. Among the latter, some continued the intervention during the first postnatal week [61,66], but the rest covered the entire gestation and the postnatal period until weaning [26,35,37,39,42]. A longer period was applied by Brandner [65] who started choline supplementation at the end of the first week of pregnancy and maintained it for two postnatal months. 

These long interventions prevent conclusions on the specific developmental processes involved in the results obtained. However, the studies focused on specific prenatal periods coincide in the duration and timing of the choline intervention since nearly all cover a 7 day period from the embryonic days (ED) 12–17, which coincides with the origin of the basal forebrain cholinergic neurons and critical changes in the hippocampal development. Early postnatal treatments that also cover hippocampal late developmental processes extending until the third week of life have been used [68].

A different picture appears with regard to postnatal interventions since they vary in duration from 14 to 270 days. In spite of this, we have identified that the most used durations are those around 30 days [30,34,44,47,49,50,62] and 90 days [38,43,63,67], but longer durations of several months have also been applied [27,42]. 

### 3.5. Behavioral Tasks and Processes Targeted

The studies we have investigated have applied a bulk of procedures for assessing cognition and emotion in combination with other processes, such as motor performance [33,50] and social interaction in rodents [37,40]. Although it is difficult to dissociate the processes involved in a given procedure and the same task might be modified in order to explore different processes, there are specialized behavioral tests such as the rotarod intended for evaluation of motor coordination and balance [73], the open-field [74], the elevated plus maze [75] and the forced swim test [76] for emotionality/anxiety and the three-chamber test for social interaction. Amongst the studies revised, although the rotarod has only been applied by Pacelli et al. [50] and the three-chamber test has only been used in a mice model of autism [37], the use of tests assessing emotionality or anxiety such as the open field or the elevated plus maze has been more frequently applied [37] and in nearly all the cases combined with cognitive tasks [28,33,46,56,61].

Regarding cognition, different cognitive processes have been assessed using a variety of learning tasks that have allowed the researchers to explore attention, learning and memory. As expected, often several studies applying the same task have been performed by the same group since each particular laboratory develops expertise and technical facilities to apply it. This favor within-laboratory comparisons but prevents comparisons between laboratories. However, the most widely extended standard tasks among labs are mazes intended for assessing spatial learning and memory. Most of the studies included in this review have assessed hippocampal-dependent declarative memory applying procedures to induce relational learning in the Morris water maze [26,27,28,46,52,58,61,62,63,64,66,69], the radial version of the water maze [39,42] and the Barnes maze [36]. Other studies assessed short-term working memory in the Morris water maze changing the location of the escape platform between sessions [52,54] and Brandner [65] modified the procedure to assess attention to the extra-maze cues. 

The novel object recognition task is also widely accepted as a test of declarative recognition memory but both the memory processes and the neural circuit involved seem to depend on the procedure applied [8,77]. Nine studies have applied the novel object recognition task to explore memory with various retention delays ranging from 20 m [44], 1 h [32,40], 6 h [32,40], 3 h [33], 24 h [29,31,34,41] to 48 h [31,41]. 

With respect to nondeclarative memory, different learning tasks are included in the studies identified. Associative learning has been assessed in classical conditioning procedures of fear conditioning [30], active avoidance [47,49], passive avoidance [67], taste aversion learning [38], context aversion [45], context-dependent conditioning and extinction [53] and visuospatial discrimination learning [68]. Instrumental learning with varying reinforcement schedules in operant chambers has been used to assess temporal processing [55,57,59,60]. Procedures to evaluate attention have been applied in both classical conditioning [38,43] and operant conditioning [35,51].

### 3.6. Effects of Dietary Choline Supplementation on Cognition

#### 3.6.1. Spatial Learning and Memory

Table 2 summarizes the results reported using the water maze task in order to assess relational spatial learning and memory. We have found reports indicating either no significant [46,62] or detrimental [61] impact of the choline supplementation on the standard water maze task with the hidden escape platform in a fixed location, several extra-maze cues and changed starting points. Amongst those studies which have focused on nonpathological samples, Glenn et al. [46] applied prenatal (ED 10–22), adolescent (PND 25–50) and adult (PND 75–125) choline supplementation in rats. When they reached adulthood the animals were trained for 3 days (4 trials per day) followed by probe trials without a platform on the last training day and a week later. An additional reversal task (4 trials) in which the platform was located in the opposite quadrant to that used in the first training followed the second probe test. No significant effects of choline supplementation on escape latencies or time searching in the target quadrant during the probe test were found in any of the groups although beneficial effects of early developmental choline supplementation were seen in emotional reactivity using the open-field and forced swim tests. Likewise, Guseva et al. [62] started choline supplementation of male rats on PND-31 and continued it for either 14 or 28 days, thus covering adolescence. They found no significant effects of supplementation on escape latencies throughout the 5 training days (4 trials per day) nor in quadrant exploration during the probe test applied 4 h after the last training trial. Moreover, a detrimental effect of choline supplementation was reported by Plyusnina et al. [61]. They trained the adult male offspring of rats fed during gestation and the first week of lactation a diet enriched with low doses of choline (1 and 0.5 g respectively) and betaine for 7 days (4 trials per day) and applied the retention probe trial on day 8. The supplemented rats exhibited slower learning than the nontreated controls with higher escape latencies and thigmotaxis on all the training days and higher search latencies during the probe trial. The authors excluded an interpretation of the learning and memory deficit in terms of increased anxiety since no significant effects were found in emotionality tests such as the elevated plus maze and the light–dark chamber. 

In fact, no significant effect of choline supplementation is the most frequently reported result in the nontreated and nontransgenic control groups included in the research with models of pathological conditions discussed below [27,28,39,52,58,63]. As expected, the beneficial effect of either prenatal or postnatal choline supplementation on the standard Morris water maze is revealed in the groups exhibiting impairment due either to immaturity or to pathological conditions. Regarding the former, one study reported that the availability of prenatal choline plays a relevant role in the developmental course of hippocampal-dependent learning and memory abilities [64]. In fact, prenatally supplemented rats were able to perform a standard noncued water maze task from 18–19 postnatal days (PND 18–19) that could not be solved by nontreated control rats until about 3 days later. The behavioral results seemed to be associated with advancement of the hippocampal development as the authors reported upregulation of the MAPK/CREB signaling cascade amongst other biochemical changes.

With respect to the value of choline supplementation to treat the deleterious effect of particular pathological conditions, spatial learning and memory improvements have been reported in the APP/PS1 mice model of Alzheimer’s disease [26,27,28], the Ts65Dn mice model of Down’s syndrome [39,42], after traumatic brain injury [58], impoverished environment [63] and seizure inducing doses of pilocarpine [69] or kainic acid [48,66]. Reestablishment of the spatial learning and memory abilities in the transgenic APP/PS1 mice model of Alzheimer’s disease has been reported after adult chronic choline supplementation from about 2 to 10–11 months of age [27,28] and early supplementation covering the prenatal and postnatal period until weaning [26]. The treated transgenic mice improved in comparison with the nonsupplemented transgenic mice and reached the level of the control nontransgenic mice both in escape latencies during the learning trials (4 per day for 5 days) and in various indexes during the retention probe trial applied on day 6. Moreover, adult choline supplementation reduced the anxiety indexes exhibited by the transgenic mice in the open field, elevated plus maze and the light–dark chamber tests [28]. Notably Velazquez et al. [26] reported a beneficial effect of early choline supplementation on spatial learning in the second generation that was never supplemented. This transgenerational effect was evident in a faster learning throughout the acquisition trials but not in the retention test. 

A modified Morris water maze consisting of a water radial maze was applied to assess spatial learning and memory in the trisomicTs65Dn mice in order to avoid the thigmotactic behavior characteristic of these mice. Using this task improved performance of adult male mice receiving dietary choline supplementation prenatally [39] or extended postnatally until weaning [42] has been reported. The beneficial effect is selectively evident in the relational task with the hidden platform for the 15 day acquisition training (5 trials per day) but not in the visible platform task which does not require memory demands.

Likewise, postnatal supplementation starting at adolescence (PND 28–30) alleviated the spatial learning and memory impairments induced by traumatic brain injury 28 days later [58] and impoverished environment after 3 months [63]. Also in a rat model of epilepsy, prenatal choline supplementation prevented the learning deficits of adolescent rats after pilocarpine [69] and kainic acid [66,78] administration in the standard relational water maze task. Similarly, choline supplementation covering the entire gestation alleviated the malnutrition-induced adolescent learning and memory deficits assessed in the Barnes maze [36]. Moreover, 4 weeks of dietary choline supplementation following the status epilepticus reestablished the spatial learning abilities [66]. In addition, using a modified version of the Morris water maze it has been reported that prenatal choline supplementation alleviated the working memory deficits induced by prenatal alcohol [52]. 

#### 3.6.2. Object Recognition Memory

The beneficial effect of choline supplementation on object recognition memory is a consistent identified finding in the eight studies carried out on rats by three different research groups. The studies carried out by the different groups are easily comparable because they applied the same choline administration procedure and doses as well as the same standard behavioral procedure that included two sessions after habituation with the experimental chamber. In the first familiarization, session two identical objects were exposed whilst in the second testing session one of the objects was substituted by a novel object. They also used similar every day and junk objects of different colors, materials and shapes. According to the rodents’ interest for novelty, object recognition was indicated by lower exploration time of the familiar object in comparison with the novel object. Either the exploration time or an exploration ratio was used as dependent variable. The main procedural difference between the groups lies in the retention period applied. Whilst de Brugada and collaborators were interested in long-term recognition memory and they applied 24–48 h delays between the familiarization and testing sessions [29,31,34,41], Kennedy et al. [32,40] used 1 and 6 h, and Glenn’s group used shorter delays ranging from 3 h [33] to 20 min [44]. 

Moreno et al. have reported that prenatal choline supplementation (ED 12–18) improves object recognition memory at long retention delays of 48 h [41], and it advances about one week the ontogenetic development of the offspring’s long-term memory abilities [29]. Younger supplemented 21–22 day old rats that were able to recognize the familiar object 24 h after the familiarization session, a task that could not be solved by nonsupplemented rats until a week later. Moreover, postnatal adult supplementation also enhanced long-term recognition memory at 48 h retention delays [31]. This effect is evident using a single 48 h retention test but not if two consecutives 24 h and 48 h tests are applied, probably because the overall level of performance hindered the detection of the supplementation effect inducing a ceiling effect. Postnatal choline supplementation covering adolescence (PND 21–60) was also able to alleviate the long-term recognition memory deficits induced by the stress induced by early maternal separation [34]. The adult rats were tested in the standard object recognition memory task and in a place recognition memory task in which the identical objects remained during the testing session but one was displaced in order to assess retention of the previous location. Whilst maternally separated, groups failed to retain for 24 h both the familiar object and the familiar location, the separated supplemented group performed at the level of the nonseparated groups.

Regarding the value of prenatal choline supplementation to alleviate pathological conditions, Kennedy et al. have reported that prenatal (ED 11–18) choline supplementation prevented the object recognition memory impairment induced by gestational iron deficiency seen in the 6 h delayed retention test [40] and postnatal (PND 11–30) choline supplementation improved performance in the 1 h delay test [32]. 

Finally, prenatal and postnatal choline supplementation has been reported to attenuate the vulnerability to the adult insult induced by the NMDA receptor antagonist MK-801 which is used to model schizophrenia-like cognitive impairments. A perinatal supplementation period (ED10-PND2) prevented the object recognition memory deficits induced by low doses of MK-801 administered after the familiarization session during the 3 h delay before the retention tests session [33]. Likewise, periadolescent (PND 25–50) choline supplementation prevented the object recognition memory impairment induced by MK-801 administration in the offspring of rats subjected to gestational stress [44]. The authors applied a standard procedure with a 20 min retention delay before and one week after MK-801 administration at the age of 75 days. While in the test prior to the insult, both maternally stressed supplemented and nonsupplemented rats were able to recognize the familiar object, but only the supplemented rats exhibited object recognition after MK-801 administration.

#### 3.6.3. Other Learning and Memory Processes

Consistent with the above results obtained in spatial and recognition memory procedures which are considered hippocampal-dependent, when we looked at the studies using other conditioning tasks we found that they showed a beneficial effect of choline supplementation in those procedures that introduce delays and context processing. This was evident in a spatial task performed in the T-maze that required associating a visual cue with the reinforced arm. The beneficial effect of early postnatal (PND 2–21) intubated choline supplementation in rats was evident when a delay between the visual cue and the access to explore the maze in all the groups was introduced. However, in absence of a delay the benefits were only detectable in the group receiving prenatal alcohol [68]. It has also been reported that prenatal choline supplementation (ED 12–17) selectively alters the contextual control of extinction without affecting associative learning or conditioned inhibition in an appetitive task [53]. Accordingly, Gitik et al. [30] reported a protective effect of a 30 day period of choline supplementation in mice against the deleterious effect of nicotine administered for 12 days during periadolescence on fear conditioning. The authors assessed the conditioned freezing response to an auditory cue and to the learning context. The beneficial effect of choline supplementation was especially evident in contextual fear conditioning, but the supplementation also enhanced cue fear learning in the group receiving nicotine during preadolescence. Similarly, a 7 week adult choline supplementation period enhanced learned context aversions induced by lithium chloride injections [45].

Likewise, time can be considered as a type of context [79]. Thus, special attention has been paid to the effect of prenatal choline supplementation on temporal processing by applying reinforcing schedules of the rat operant responses based on temporal criteria. The peak-interval timing procedure applied by the research group focused on this issue is a variant of the fixed-interval procedure in which the reward is delivered a fixed time after a visual or auditory signal stimulus disappears when the response takes place. The introduction of nonreinforced trials and changes in the duration of the signal stimulus allows the researcher to record the entire response (initiation and termination) including changes not only of the response rate but the response timing which reflects the time perception [60]. They have also used the differential reinforcement of low-rate schedules in which the response is only reinforced if it takes place after a minimum amount of time following the previous lever press [59]. In this case, prior responses are not reinforced, but instead the time is reset [59] so that it requires regulating the response to be regulated according to time cues. Several reports have confirmed that prenatal choline supplementation (ED 12–17) increases the sensitivity to the signal duration and the precision in the temporal control of response in adult [60] and aged rats [55]. Sex differences have also been described as choline supplementation seemed to improve performance by facilitating behavioral inhibition in females but to impair it in males [60]. Furthermore, choline supplementation enhances the ability to reset the clock after a nonreinforced gap and the sensitivity to added sensory information during these periods [57]. 

In addition, a modest benefit of dietary choline supplementation for 28 days [49] and no effect of supplementation for 14 days [47] have been reported in the active avoidance learning performance of rats subjected to neurotoxic insult by the administration of the cholinesterase inhibitor soman. Regarding taste aversion learning, choline supplementation for 12 weeks has been found to slow down extinction, thus suggesting learning enhancement [38].

#### 3.6.4. Attentional Processes

Based on the proposed role of acetylcholine in processing relevant stimuli and suppressing nonrelevant cues [13], different studies have assessed the effect of choline supplementation on attention using various learning and memory tasks. However, the results are often of difficult to interpret. 

On the one hand, Brander [65] reported improved “buffering” or inhibition of salient cues in a version of the Morris water maze task which included a salient suspended cue signaling the platform location in addition to other extra-maze cues in the room. The fact that the local cue indicated the fixed platform location during training prompted a nonrelational spatial learning strategy based on the attention attracted by the local cue. Thus, changing the location of the cue to an irrelevant location or removing it during a probe trial without a platform impaired the performance in nontreated control groups. However, those rats that received choline supplementation from the second postnatal week to the second postnatal month exhibited a better performance at the age of 5 months in the probe test. Contrary to the control rats that searched in the location signaled by the cue, the treated rats searched for longer in the previous platform location. This can be interpreted as a better relational memory based on all the available cues thanks to the inhibition of attention to the salient local cue or to a difficulty in attending to the relevant cue. In any case, it shows that dietary choline availability during early development induces long-term modification of the attentional processes.

On the other hand, research using the “latent inhibition” phenomenon to explore the effect of choline supplementation on attention has yielded opposite results in rats depending on the learning procedure applied. Latent inhibition is induced by the prior exposure without consequences to the conditioned stimulus used in a later learning. The pre-exposure produces a retard in the acquisition of learning which has been explained in terms of attentional changes as the stimulus has become irrelevant. An identical 3 month postnatal choline supplementation which disrupted latent inhibition of the conditioned emotional response [43] had no effect of latent inhibition of taste aversion learning [38]. The authors explain this discrepancy in terms of different brain circuits involved in the different learning tasks with varying contributions of cholinergic innervation. 

Finally, a visual learning task in which the nose-poke response in one of five ports signaled by a discriminative visual cue triggered the presentation of reward was used to explore the effect of choline supplementation applied from gestation to weaning on attention in adult Ts65Dn mice [35,51]. The authors varied parameters such as the duration of the cue and the delay prior to the cue presentation in order to increase attention demands. They reported beneficial effects of supplementation both in trisomic and disomic control mice with shorter durations without affecting the simple discrimination task. The authors were even able to dissociate the domains affected. Supplemented disomic controls reduced the number of incorrect responses suggesting increased vigilance and supplemented Ts65Dn mice showed improved performance after a previous error by reducing the emotional response. 

Therefore, the revised results of the studies point to the relevance of choline availability in regulating attention, but more research is needed to understand the nature of its role.

### 3.7. Effects of Dietary Choline Deficiency on Cognition

Eleven studies used choline deficient groups subjected to a 0% choline diet [34,38,41,44,50,53,54,57,62,66,67]. Four of them were performed with a subject with a prior pathological condition such as maternal separation [34], prenatal stress [44] and induced seizures [66,80]. Below, we review the effects of dietary choline deficiency based on the behavioral-cognitive processes evaluated.

On the one hand, the Morris water maze was the most commonly used task to assess the effect of dietary choline deficiency on spatial learning and memory. Amongst the three studies that used this task, two of them reported no effect of dietary choline deficits. No effect was found in young-adult rats after deficiency periods of 14 or 28 days starting on PND 31 [62]. Neither did prenatal deprivation induce any effect in adults and after the status epilepticus induced with KA as the deficient groups exhibited a similar performance to that of the control group fed a standard 1.1 g/kg choline diet [66]. However, Wong-Goodrich et al. [54] reported that rats subjected to a 12 week dietary choline-deficiency period were impaired on a matching-to-place water-maze task assessing short-term working memory. Interestingly, in another experiment the authors also used a 12-arm radial-maze task in which the animals learned the location of the 8 baited arms that were held constant throughout the experiment. The dietary intervention was more complex in this experiment because it combined prenatal and postnatal dietary interventions. They reported that the adult rats that received the control diet showed no effect of adult choline deprivation and prenatal choline deficiency induced no effect on adult rats fed the control diet. However, prenatal choline deficiency together with adult supplementation did impair acquisition. A similar result was obtained with a mismatch in choline content between prenatal supplementation and adult choline privation. The authors indicated that these results point to prenatal choline intake as a major factor in determining the level of choline intake that is needed in adulthood for optimum spatial memory performance. Moreover, this hypothesis could explain the negative results obtained in the two Morris water-maze studies previously discussed.

On the other hand, spatial learning and memory has also been assessed in place recognition memory tasks. Thus, Moreno et al. [34] found that rats feed with a choline deficient diet during periadolescence (PND 21–60) showed difficulties in remembering the object localization in a 24 h test. However, they did not find any difference in performance between deficient and control groups neither in object recognition memory or place recognition memory tasks when these groups suffered a maternal separation since both groups were unable to remember the object and place in the test. A memory failure in object recognition after prenatal choline deficiency has been reported when the test is performed at 48 h but not at 24 h [43]. However, the detrimental effect of prenatal choline deprivation can emerge even at short intervals of 20 min if stressful or challenging situations occur. Thus, while the control choline groups only showed this memory deficit after combined prenatal stress and adult neurotoxic administration of the NMDA receptor antagonist MK-801, the choline-deprived groups exhibited this memory failure with the sole presence of just one of these events [44]. Again, these results indicate that the dietary choline deprivation could be a vulnerability factor for future aversive situations or that these deficits only emerge in especially difficult or demanding tasks.

Furthermore, there are other cognitive processes affected by dietary choline deprivation. Auditory information can be used as a relevant context to determine the performance of previous learning responses. Thus, Lamoureux et al. [53] and Buhusi et al. [57] found that prenatal choline deprivation could impair processing of contextual sensory context information. Deficient groups failed to exhibit sensitivity to a context change both in the renewal of an excitatory classic conditioning task and an instrumental task, respectively. Moreover, choline deprivation in adults also impairs the effect of previous exposures on later taste aversion learning [38]. Adult rats fed for 12 weeks with a deficient choline diet were unable to show latent inhibition. A similar dietary intervention also impaired retention in a passive avoidance task as choline-deprived rats required fewer trials to enter the shock compartment during the extinction phase [67]. 

### 3.8. Assessment of the Dietary Choline Effects on Emotionality/Anxiety

Although the search terms of this review focused on those studies aimed at evaluating the effect of dietary choline availability on cognition, some of the identified studies assessed emotionality/anxiety in order to explore different behavioral domains. Various tests have been used in the studies reviewed. The open-field test consisting of an empty arena has been used to assess anxiety-like behaviors such as low exploratory locomotion and avoidance of the center area thus reducing the exploration to the area close to the walls, a behavioral pattern called thigmotaxis [74]. The elevated plus maze is also a behavioral test of anxiety. Anxious rodents avoid the two open arms in comparison with the two enclosed arms, thus exhibiting a low number of entries and time spent in them [75]. Lastly, the forced swim test assesses behavioral despair as the animal is placed in a container full of water [76]. Since it represents an inescapable situation, low latency to cease attempts to escape and immobility are considered indexes of depressive-like behavior. Other less frequent measures of anxiety are marble burying and the light–dark test.

The only study identified which included both cognitive and emotionality/anxiety tests in a nonpathological sample of female rats explored the effect of prenatal (ED 10–22), adolescent (PND 25–50) and adult (PND 75–125) choline supplementation [46]. Whilst no effect was seen in cognitive performance assessed in the Morris water maze, early supplementation decreased anxiety-like behavior in the open-field and behavioral despair in the forced swim test. Thus, adult rats supplemented either prenatally or during adolescence but not during adulthood increased exploration and time spent in the center of the open-field test while prenatal supplementation increased mobility in the forced swim test. Accordingly, prenatal choline supplementation (ED 12–17) attenuated, but did not prevent, the age-related decline in exploration of the open field [56]. This is consistent with a beneficial effect of a 9 month adult period of choline supplementation in the open field test, the elevated plus maze and the light–dark test reported in the model of age-related dementia APP/PS1 mice [28]. However, no effect of a long choline supplementation period covering gestation and lactation was reported using the BTBR model of autism in spite of a beneficial effect in marble digging and social interaction [37]. Moreover, postnatal choline deprivation for 28 days in adult male rats did not affect performance in the open field test [50].

## 4. Discussion

A general overview of the results reported by the rodent studies reviewed confirms a beneficial effect of choline supplementation which is more evident in those tasks requiring long-term relational memory and those conditions associated with immature or impaired cognitive functions. Interestingly, except in one case, no effect of choline supplementation on the standard Morris water-maze task has been reported in nonpathological conditions (Table 2), whilst enhanced performance in recognition memory tasks was obtained in studies performed by different laboratories with different retention intervals (Table 3). Nonetheless, the improvement of spatial learning and memory arises in supplemented rodent models of pathological conditions, such as the APP/PS1 mice model of Alzheimer’s disease [26,27,28], the Ts65Dn mice model of Down’s syndrome [39,42], traumatic brain injury [58], impoverished environment [63] and seizure inducing doses of pilocarpine [69] or kainic acid [48,66]. Prenatal choline supplementation also accelerated the development of both the ability to perform the relational water maze task [64] and to recognize a familiar object after a 24 h delay [29]. Consistent with a selective beneficial effect of choline supplementation on learning tasks involving relational memory, in most of the cases, the improvement is also found in other relational learning tasks that include contextual and temporal information such as context-dependent extinction [53], contextual fear conditioning [30], context aversion [45] and peak-interval timing [55,57,60]. On the contrary, no effect has been found in appetitive associative learning and conditioned inhibition [53], auditory fear conditioning [30] and active avoidance learning [47]. This does not support our hypothesis because this is often interpreted as a selective effect of choline supplementation on learning tasks requiring hippocampal-dependent relational memory. However, the selective effect of choline supplementation on specific tasks can be explained either by the level of task difficulty so that a ceiling effect could mask the impact of supplementation in the easier tasks or by a selective effect of supplementation on specific brain systems. Regarding the former, it is not easy to assess the level of difficulty of a particular task, and it seems puzzling that a simple recognition of an object after a short retention interval could be more difficult than locating a noncued hidden platform on the basis of a relational spatial map. Nevertheless, the second task is less affected by dietary choline interventions (Table 2 and Table 3). Based on the results reported by the revised studies, it is also difficult to draw conclusions on the latter issue. Although the evidence is scarce, some results point to a more general beneficial effect of dietary choline on emotional processes. 

Only 16 of the 44 studies reviewed included assessment of the brain changes induced by dietary choline supplementation and were mainly focused on the cholinergic neurotransmission and the hippocampus. Several outcomes of choline supplementation were reported, such as increased levels of brain choline [36], enhanced hippocampal neurogenesis [33,48,54,56], upregulation of α7 nicotinic acetylcholine receptors in cortex and hippocampus [58,62] and increased activation of hippocampal MAPK (mitogen-activated protein kinase) and CREB (cAMP-response element binding protein) [64]. Choline supplementation that improved performance in the water maze, attentional and emotionality/anxiety tasks partially prevented the reduced number and abnormalities of basal forebrain cholinergic neurons, increased the amygdala and hippocampal cholinergic innervation and ameliorated amyloid pathology in APP/PS1 mice correlating the brain changes with better behavioral performance [26,27,28,35]. Similarly, the increase in basal forebrain cholinergic neurons [39] and hippocampal neurogenesis [42] in supplemented Ts65Dn mice has been reported to correlate with spatial learning abilities. Likewise, choline supplementation ameliorated the contextual fear-conditioning deficits and reversed the methylation of 462 genes induced by adolescent nicotine exposure in the adult dorsal hippocampus [30].

The current theoretical framework identifying cognition with conscious declarative/relational memory determines a tendency to focus on the changes induced by dietary choline interventions on the hippocampus and hippocampal-dependent learning tasks. Given this focus, the general available evidence supports a beneficial role on relational learning and memory. However, cognition is a wide term covering a variety of learning processes closely related with attention and emotion which depend on widely distributed brain systems. Thus, taking into account the available results, a more general impact of choline supplementation cannot be excluded. As a matter of fact, although scarce, some studies have found evidence supporting the fact that choline supplementation effects extend not only to other learning [38,49] but also to attentional [38,43,65] and emotional tasks, such as the open field and the elevated plus mazes, amongst others [28,37,56]. Moreover, improved performance in the open field and forced swim tests after prenatal and adolescent choline supplementation in females that did not affect learning in the water maze [46] has also been reported. Furthermore, considering only the reported impact of dietary choline on the cholinergic system, widely spread brain effects of choline intervention are consistent with the wide-ranging diffuse cortical and subcortical projections of the basal forebrain system as well as the extensive interactions with other neurotransmitters. This is supported by the limited number of studies that report effects of dietary choline on nonhippocampal brain areas such as the cortex [62] and amygdala [28]. In addition, other potential effects of dietary choline on brain function remain widely unexplored. Only one study has described epigenetic changes associated with cognitive enhancement induced by choline supplementation [30].

With respect to a potential sex-dependent differential effect of choline supplementation, the results contained in the studies identified are scarce, since only eight of them included both males and females and one of them did not specify the statistics regarding the factor “sex” [26]. In most of the studies, sex did not influence the basal behavioral performance nor did it have a significant impact on the effectiveness of choline supplementation so that the data of both sexes data collapsed [37,52,64,65,68]. However, the low number of subjects (4–5 of the same sex per group) might have hindered obtaining significant differences. Accordingly, the only two studies using a higher number of subjects (8 per group of the same sex) reported sexually dimorphic performance and supplementation effects. Firstly, in an operant learning task in which reinforcement is delivered following a differential reinforcement of low-rate schedule with changing temporal criteria female rats were more efficient than male rats and prenatal choline supplementation increased the number of rewarded responses in females but not in males [59]. Although both supplemented males and females reduced impulsive responses at very short delays, supplemented males had a higher number of nonrewarded responses prior to the correct time interval, a behavior interpreted by the authors as resistance to changes of the temporal criteria. Therefore, supplementation increased the response efficiency in females and decreased it in males. This can be attributed to a beneficial effect of the supplementation on the emotional behavior regulation related with impulsivity and frustration in female rats which can be related with gonadal hormones. Secondly, different effects of prenatal choline supplementation have been reported in both male and female rats in emotionality/anxiety tasks using prepubescent (1 month old) and aged (24 month old) samples which were chosen in order to eliminate the influence of gonadal hormones [56]. Aging decreased to a greater extent the exploratory behavior in the open field of males in comparison with the females, and supplementation attenuated the age-related deficit in the males but not in the females. On the contrary, supplementation tended to increase object exploration in adolescent females but not in adolescent males. Although the age of the samples chosen corresponds to periods of low gonadal levels, young females exhibited higher levels of corticosterone than young males which were larger in response to restrain-induced stress. Thus, differences in the anxiety responses to novel environments or objects influenced by gonadal hormones can explain the differential effects of prenatal choline supplementation on exploratory behavior in males and females at different stages of the lifespan. Accordingly, prenatal and adolescent choline supplementation improves the performance of female rats in emotionality/anxiety tests but not in spatial learning [46]. Therefore, in spite of the scarce evidence available, the modulation of the dietary choline effect by sex merits attention since it seems plausible to propose a more relevant role of choline in the regulation of emotionality in females. 

Regarding the optimal supplementation periods, there is general agreement in the relevance of the prenatal period ED 12–17 coincident with neurogenesis in the basal forebrain cholinergic nuclei. Fourteen reports found beneficial effects of supplementation during this period in recognition memory [29,33,40,41], spatial learning and memory [48,64,69], working memory [54], context-dependent conditioning and extinction [53], temporal processing [55,57,59,60] and anxiety [56]. The long-term effects extend throughout the lifespan for months reaching advanced ages such as 20 [54] and 24 months of age [56] and even the following nontreated second generation [26]. Moreover, independent laboratories have found advanced maturation of learning abilities after this supplementation period in young rats [29,64]. Therefore, the effects reported by the rest of studies that applied longer prenatal and perinatal supplementation could be due the choline impact on this critical period cannot be ruled out since all of them covered it.

A different picture arises with respect to postnatal supplementation as the varied durations of the dietary interventions make comparisons difficult and prevented us from drawing general conclusions. Taken together, the results do not provide a clear support for a significant effect on cognition. The most commonly used choline supplementation period of approximately 30 days yields different outcomes. Guseva et al. [62] did not find a significant effect of either 14 or 28 days of supplementation on spatial learning and memory in spite of using a diet that contained seven times the choline levels of the standard diet and induced nicotinic receptor upregulation in the cortex and hippocampus. Similarly, a supplemented choline diet containing seven times as much choline as the standard diet for 28 days did not protect of the detrimental effect of the toxicity induced by exposure to soman in active avoidance [47,49]. A 28 day deprivation period impairing motor coordination also caused modest impairment in active avoidance but no effect in passive avoidance and open field tasks [50]. However, the administration of the choline supplemented standard diet for a period of 25 days after weaning (PND25–50) was enough to mitigate the object recognition memory impairment induced by prenatal stress and by the NMDA antagonist MK-801 used as a model of schizophrenia in rats [44]. Gitik et al. [30] also reported reversal of nicotine-induced deficits in contextual fear conditioning after a 30 day supplementation period in mice, and Moreno et al. [34] found increased long-term recognition memory after a 39 day supplementation period in rats. In all these cases, the supplementation took place in very young animals, and the behavioral effect was acute as the animals were tested either during the supplementation or immediately after. Longer supplementation treatments of several months in adulthood are scarce and yield inconsistent results in attentional measures [38,43]. At the same time, supplementation for periods around two months enhance learned context aversions [43] and long-term retention in recognition memory [31] whilst three months of choline supplementation but not shorter periods did reverse the spatial learning impairment in rats reared in impoverished environment [63]. It is noticeable that choline supplementation periods as long as 7.5 [27] and 9 months [28] ameliorating the cognitive and anxiety deficits in the transgenic APP/PSI mice did not even induce improvement in the control nontransgenic mice. Therefore, the results of research applying postnatal interventions in nonpathological models are not conclusive probably due to ceiling effects in contrast to the more clearly supported beneficial effect of long supplementation periods to reverse the detrimental cognitive effects associated with pathological conditions. In addition, the interaction between the level of prenatal and postnatal choline availability should be taken into account since there are results indicating that switching either from a deficient prenatal diet to a supplemented postnatal diet or from a prenatal supplemented diet to a postnatal deficient diet has detrimental effects on spatial learning in the 12-arm radial maze [54]. 

All in all, the results reviewed provide compelling support for the relevance of prenatal dietary choline in regulating the cognitive functions of offspring. More research is needed to draw conclusions about the effectiveness of postnatal dietary choline interventions since the limited number of studies that have been carried out exhibits great variability regarding the treatment length, behavioral tasks and outcomes. Positive effects described mainly in restoring functions impaired by pathological conditions presents this endeavor as worthy. The possibility of establishing comparisons between tasks is missing in most of the reports reviewed. Given the fact that different behavioral tasks assess different attentional, emotional and cognitive processes which depend on dissociable brain systems, a careful selection of behavioral tasks to precisely evaluate the nature of the dietary choline interventions in cognition as well as dissociating the effect on attentional, activational, motivational and emotional processes would be required. In spite of the fact that the learning and memory results point to a selective effect on relational tasks, a more general effect on other processes cannot be ruled out given the positive findings in other tasks. It would be advisable to develop behavioral test batteries to be applied in independent laboratories, although this may be difficult because it depends on the expertise and facilities available to each group. This would permit a systematic assessment of variables such as timing, duration and doses which is lacking at present. A deeper knowledge of the impact in cognition would also help to lead research on the brain mechanisms involved in the effect of dietary choline interventions beyond the hippocampus. Even though the hippocampus is critically involved in the acquisition of relational learning and long-term memory, it forms part of distributed brain networks involving cortical and subcortical brain areas. It is striking that in nonpathological conditions, dietary choline supplementation induces a consistent improvement of recognition memory tasks which require the perirhinal cortex integrity [8] to a greater extent than the hippocampal integrity [77], whilst the effect is absent in hippocampal-dependent water maze tasks according to the reviewed reports. Therefore, a wider scope in the investigation of the brain functions underlying the effect of dietary choline interventions in cognition is desirable. 

To sum up, the results of the reviewed research indicate an influence of the dietary choline in cognition, but further research is needed to identify the critical variables and the nature of the impact on specific processes. It seems advisable to focus on adult choline supply by establishing comparisons in the same study between the effects of several dosages and durations of the intervention period on various behavioral tasks reflecting different cognitive processes. Likewise, including larger groups of females is required as sex-dependent effects can be expected. Furthermore, it would be worthwhile to extend the investigation of the physiological mechanisms involved in the behavioral effect of dietary choline beyond the cholinergic neurotransmission in order to concentrate on the phospholipid metabolism and epigenetic changes proposed as relevant for cognition not only during early development but also adulthood and aging [20].

The results obtained in rodents do not permit us to draw conclusions regarding the value of dietary choline interventions in humans, but they point to critical issues to be considered in human studies such as choline dosage, timing and duration of the intervention, as well as the type of cognitive assessment applied. The finding that dietary choline has been shown to be more efficient in pathological conditions suggests the possibility of easy and noninvasive interventions which deserve to be assessed in clinical trials.

## Figures and Tables

**Figure 1 nutrients-13-01966-f001:**
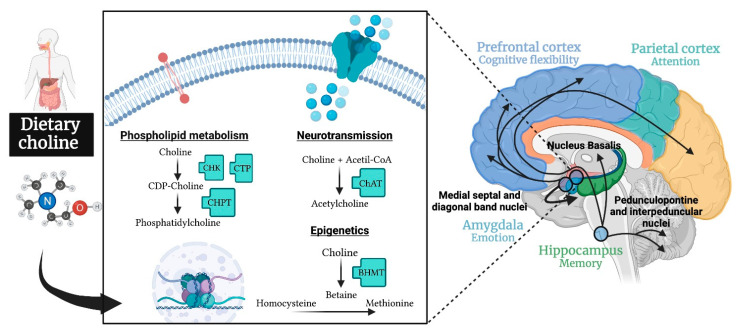
Schematic diagram depicting the biological pathways by which dietary choline supply can influence the brain function related to cognition. Most of the available evidence focuses on the role of choline as a precursor of the widespread neurotransmitter acetylcholine associated with the modulation of brain areas related with attention, memory, behavioral flexibility and emotion. Emerging evidence highlights the potential relevance of its contribution to maintain cell membranes through the phospholipid synthesis and to the epigenetic changes associated with cognition as a methyl donor. Abbreviations: BHMT, betaine homocysteine-methyltransferase; CDP-choline, cytidine 5′-diphosphocholine; ChAT, choline acetyltransferase; CHK, choline kinase; CHPT cholinephosphotransferase; CTP, phosphocholine cytidylyltransferase. Created with BioRender.com.

**Figure 2 nutrients-13-01966-f002:**
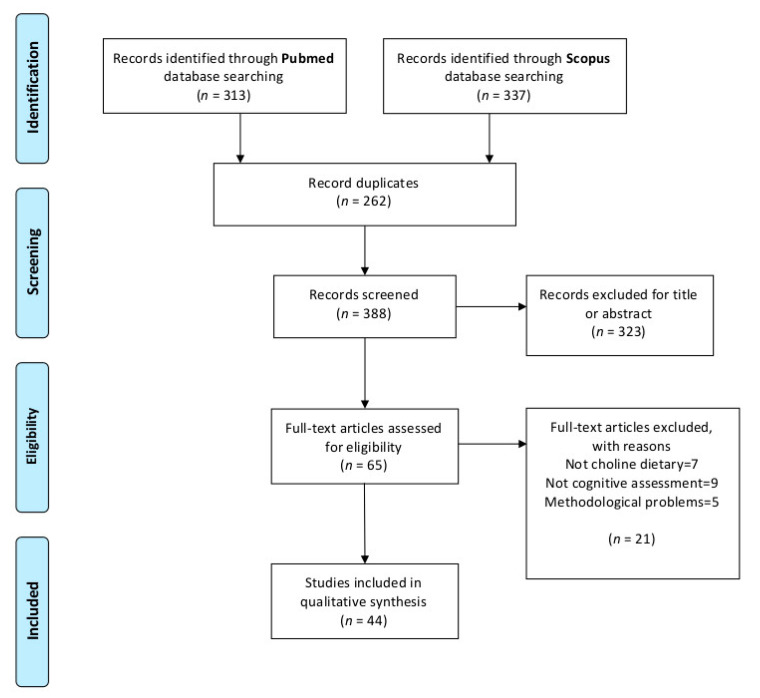
PRISMA (Preferred Reporting Items for Systematic Reviews and Meta-Analyses) flow diagram of papers selection for inclusion in the review.

**Figure 3 nutrients-13-01966-f003:**
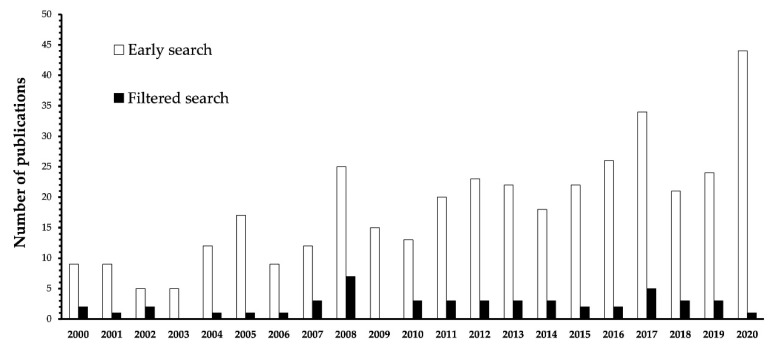
Number of articles published by years on the topic. White bars show the raw results found in the initial databases search, and black bars the final selection included in this review.

**Figure 4 nutrients-13-01966-f004:**
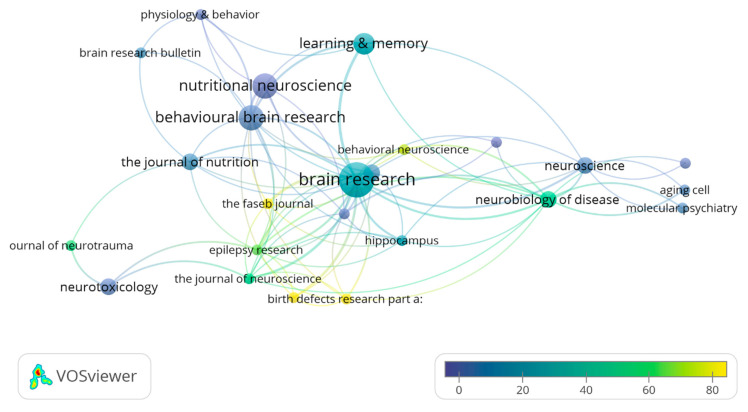
Network data map of the scientific journals in which the articles have been published. Point size represents the number of published articles. The colors represent the number of citations of each article. The lines indicated the links between journals based on crosscitations.

**Figure 5 nutrients-13-01966-f005:**
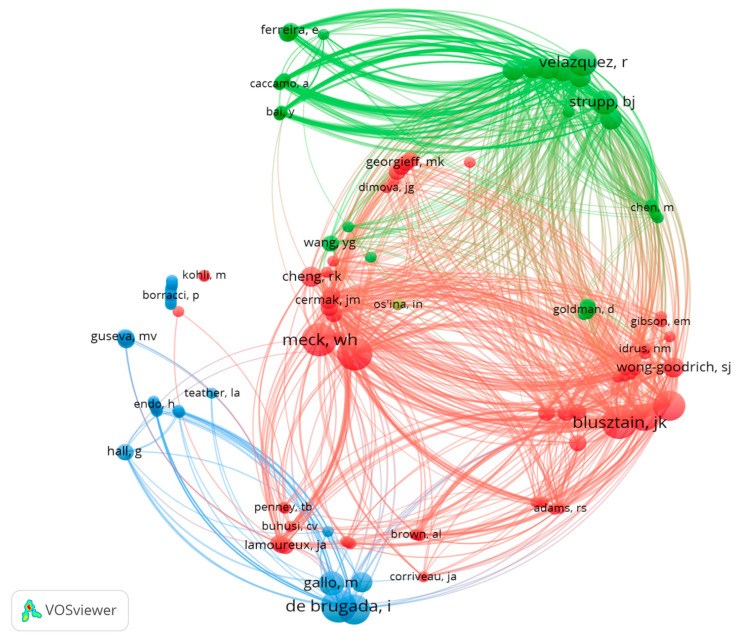
Network data map of the articles’ authorship indicating research groups and collaborations. Point size represents the number published articles. The colors and lines show clusters based on coauthorship.

**Figure 6 nutrients-13-01966-f006:**
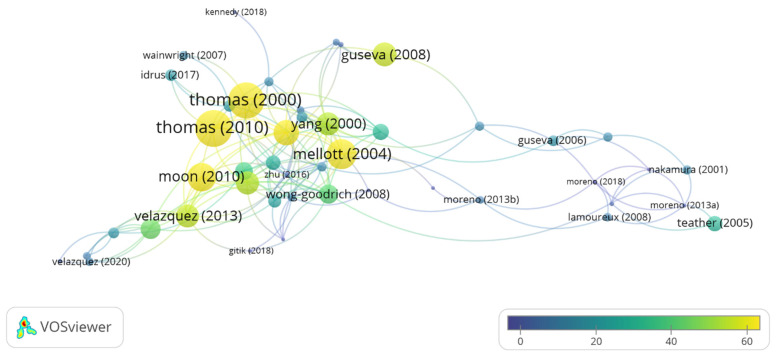
Network data map of the articles’ citations. The point size and color represent the number of citations received. The lines represent the number of co-citations.

**Table 1 nutrients-13-01966-t001:** Summary table of included studies. Supplemented (S), control (C), deficient (D), prenatal–postnatal supplementation (PRE/POST), improvement (↑), impairment (↓), no effect (=), days (d), embryonic day (ED) and postnatal day (PND).

	Choline Treatment Groups	Prenatal or Postnatal	Task Applied	Pathological Pre-Condition	Diet Effect	Treatment Duration	Subject	Sex
*Velazquez et al., 2020 [26]*	S-C	PRE POST	Morris Water Maze	Alzheimer	S - ↑	42 d (ED1-PND21)	Mice APP/PS1	Males and females
*Velazquez et al., 2019 [27]*	S-C	POST	Morris Water Maze	Alzheimer	S - ↑	225 d (PND75- PND300)	Mice APP/PS1	Females
*Wang et al., 2019 [28]*	S-C	POST	Morris water maze, Open-field, Elevated plus maze, lightdark box test and nest building.	Alzheimer	S - ↑	270 d (PND60-PND330)	Mice APP/PS1	Males
*Moreno and de Brugada, 2019 [29]*	S-C	PRE	Object recognition		S - ↑	7 d (ED12-ED18)	Rat Wistar	Males
*Gitik et al.,2018 [30]*	S-C	POST	Fear conditioned	Nicotina administration	S - ↑	30 d (PND35/PND50/PND66)	Mice C57BL/6J	Males
*Moreno et al., 2018 [31]*	S-C	POST	Object recognition		S - ↑	70 d (PND180-PND250)	Rat Wistar/Hooded Lister	Males
*Kennedy et al., 2018 [32]*	S-C	POST	Object recognition	Prenatal iron deficit	S - ↑	19 d (PND11-PND30)	Rat Sprague-Dawley	Males
*Nickerson et al., 2017 [33]*	S-C	PRE	Object recognition and Open-field	Neurotoxin (MK-801)	S - ↑	13 d (ED10-PND2)	Rat Sprague-Dawley	Males
*Moreno et al., 2017 [34]*	S-C-D	POST	Object and place recognition	Maternal separation	S - ↑ D - ↓	39 d (PND 21-PND60)	Rat Wistar	Males
*Powers et al., 2017 [35]*	S-C	PRE POST	Operant attention tasks	Down’s syndrome	S - ↑	42 d (ED1-PND21)	Mice Ts65Dn	Males
*Zhu et al., 2016 [36]*	S-C	PRE	Barnes maze	Prenatal low protein intake	S - ↑	21 d (ED0-ED21)	Rat Sprague-Dawley	?
*Langley et al., 2015 [37]*	S-C	PRE POST	Open field, Elevated plus maze, Marble burying and three-chamber social interaction tests	Autism spectrum disorder	S - ↑	42 d (ED1-PND21)	Mice BTBR T+ (ASD model)	Males and females
*Gámiz et al., 2015 [38]*	S-C-D	POST	Latent inhibition of conditioned taste aversion		S - ↑ D - ↓	90 días (adults)	Rat Wistar	Males
*Ash et al., 2014 [39]*	S-C	PRE POST	Radial water maze	Down’s syndrome	S - ↑	42 d (ED1-PND21)	Mice Ts65Dn	Males
*Kennedy et al., 2014 [40]*	S-C	PRE	Object recognition and Social approach task	Prenatal iron deficiency	S - ↑	8 días (ED11-ED18)	Rat Sprague-Dawley	Males
*Moreno et al., 2013 [41]*	S-C-D	PRE	Object recognition		S - ↑	7 días (ED12-ED18)	Rat Wistar	Males
*Velazquez et al., 2013 [42]*	S-C	PRE POST	Radial water maze	Down’s syndrome	S - ↑	42 d (ED1-PND21)	Mice Ts65Dn	Males
*Moreno et al., 2013 [43]*	S-C	POST	Hall-Pearce negative transfer and Latent inhibition (attention process)		S - ↑	84 d (PND240-PND344)	Rat Wistar/Hooded Lister	Males
*Corriveau et al., 2012 [44]*	S-C-D	POST	Object recognition	Prenatal stress and neurotoxin in adulthood (MK-801)	S - ↑ D - ↓	25 d (PND25-50)	Rat Long-Evans	Males
*Moreno et al., 2012 [45]*	S-C	POST	Context taste aversion		S - ↑	49 d (PND90-PND139)	Rat Wistar	Males
*Glenn et al., 2012 [46]*	S-C	PRE PRO	Morris water maze, Open field, forced swimming		S - ↑	12 d (ED10-ED22) / 25 d (PND25-50)/ +25, (PND75- end study)	Rat Sprague-Dawley	Females
*Langston and Myers, 2011 [47]*	S	POST	Active avoidance test	Exposure to neurotoxic (soman)	S - =	28 d ( >175-200gr)	Rat Sprague-Dawley	Males
*Wong-Goodrich et al., 2011 [48]*	S-C	PRE	Morris water maze	Seizured induce by kainic acid	S - ↑	6 días (ED12-ED17)	Rat Sprague-Dawley	Males
*Myers and Langston, 2011 [49]*	S	POST	Active avoidance test	Exposure to neurotoxic (soman)	S - =	28 d ( >175-200gr)	Rat Sprague-Dawley	Males
*Pacelli et al., 2010 [50]*	C-D.	POST	Rotarod, Active and Passive avoidance test and Open-field	.	D - ↓	28 días	Rat Wistar	Males
*Moon et al., 2010 [51]*	S-C	PRE	5-choice hole operant chamber	Down’s syndrome	S - ↑	21 d (ED0-ED21)	C57Bl/6J xC3H/HeSnJ and Mice Ts65Dn	Males
*Thomas et al., 2010 [52]*	S-C	PRE	Morris water maze, Spatial working memory, Spontaneous alternation and parallel bar motor coordination	Prenatal ethanol	S - ↑	15 d (ED5-ED20)	Rat Sprague-Dawley	Males and females
*Lamoureux et al., 2008 [53]*	S-C-D	PRE	Context dependent and extinction operant chamber		S - ↓ D - ↓	8 d (ED12-ED17)	Rat Sprague-Dawley	Males
*Wong-Goodrich et al., 2008 [54]*	S-C-D	PRE POST	Morris water maze and radial maze	.	S - ↑↓ D - ↑↓	8 d (ED12-ED17)	Rat Sprague-Dawley	Males
*Cheng et al., 2008 [55]*	S-C	PRE	Auditory and visual bisection procedure	.	S - ↑	8 d (ED12-ED17)	Rat Sprague-Dawley	Males
*Glenn et al., 2008 [56]*	S-C	PRE	Open-field		S - ↑	8 d (ED12-ED17)	Rat Sprague-Dawley	Males and females
*Buhusi et al., 2008 [57]*	S-C-D	PRE	Contextual processing in operant task		S - ↑ D - ↓	8 d (ED12-ED17)	Rat Sprague-Dawley	Males
*Guseva et al., 2008 [58]*	S-C	POST	Morris water maze	Traumatic brain injury	S - ↑	>30 d (PND28-PND58)	Rat Sprague-Dawley	Males
*Cheng et al., 2008 [59]*	S-C	PRE	Operant tasks with differential reinforcement of low-rate.	.	S - ↑	8 d (ED12-ED17)	Rat Sprague-Dawley	Males and females
*Cheng and Meck, 2007 [60]*	S-C	PRE	Operant tasks with temporal manipulation)	.	S - ↑	7 d (ED11-ED17)	Rat Sprague-Dawley	Males
*Plyusnina et al., 2007 [61]*	S-C	PRE POST	Morris water maze, light-dark chamber and elevated plus test	.	S - ↓	28 d (ED0-PND7)	Rat gris agresiva	Males
*Guseva et al., 2006 [62]*	S-C-D	POST	Morris water maze		S - =	14 d (PND31-45)/ 28 d (PND31-PND59)	Rat Sprague-Dawley	Males
*Teather and Wurtman, 2005 [63]*	S-C	POST	Morris water maze	Impoverished environmental	S - ↑	90 d (PND30-120) / 30d (PND30-PND60 or PND90-PND120)	Rat Sprague-Dawley	Males
*Mellott et al., 2004 [64]*	S-C-D	PRE	Morris water maze		S - ↑	8 d (ED11-ED18)	Rat Sprague-Dawley	Males and females
*Brandner, 2002 [65]*	S	PRE POST	Morris water maze		S - ↑	14 d (ED7-ED21) + 28 d (PND1-PND28)	Rat PVG	Males and females
*Holmes et al., 2002 [66]*	S-C-D	PRE POST	Morris water maze	Seizure induced by kainic acid	S - ↑	18 d (ED11-PND7) / 28 d (PND36-PND63)	Rat Sprague-Dawley	Males
*Nakamura et al.,2001 [67]*	C-D	POST	Passive avoidance test		D - ↓	84 d (PND63-PND147)	Rat Wistar	Males
*Thomas et al., 2000 [68]*	S-C	POST	Visuospatial discrimination in T-maze	Prenatal ethanol	S - ↑	19 d (PND2-PND21)	Rat Sprague-Dawley	Males and females
*Yang et al., 2000 [69]*	S-C	PRE	Morris water maze	Seizure induced by Pilocarpina	S - ↑	7 d (ED11-ED17)	Rat Sprague-Dawley	Males

**Table 2 nutrients-13-01966-t002:** Articles that used the Morris water maze to assess the dietary choline effect. Prenatal–postnatal supplementation (PRE/POST), animal model of pathology (YES/NO) and treatment outcome (improvement/impairment/no effect). Note that the control groups without pathological conditions included in studies using pathological conditions are also considered. Pathological conditions include chronic animal models of pathologies, acute insults and deficiencies caused by prenatal/postnatal treatments and immaturity.

Reference	Pre-Postnatal		Pathology	Outcome
[26,48,52,64,66,69,78]	PRE		YES	IMPROVEMENT
[54]	PRE		NO	IMPROVEMENT
[28,48,69]	PRE		NO	NO EFFECT
[54,61]	PRE		NO	IMPAIRMENT
[39,46]	PRE	POST	NO	NO EFFECT
[39,42]	PRE	POST	YES	IMPROVEMENT
[27,28,58,63]		POST	YES	IMPROVEMENT
[27,28,46,52,58,62,63]		POST	NO	NO EFFECT

**Table 3 nutrients-13-01966-t003:** Articles that used the object recognition task to assess the dietary choline effect. Prenatal–postnatal supplementation (PRE/POST), animal model of pathology (YES/NO), treatment outcome (improvement/impairment/no effect) and retention delay between the familiarization and testing sessions. Note that the control groups without pathological conditions included in studies using pathological conditions has also been considered. Pathological conditions include chronic animal models of pathologies, acute insult and deficiencies caused by prenatal/postnatal treatments and immaturity.

Reference	Pre-Postnatal		Pathology	Outcome	Delay
[29,33,40]	PRE		YES	IMPROVEMENT	24/3 h/ 6 h
[41]	PRE		NO	IMPROVEMENT	24 h
[40]	PRE		NO	NO EFFECT	1–6 h
[32,34,44]		POST	YES	IMPROVEMENT	20 m/1 h/24 h
[31,44]		POST	NO	IMPROVEMENT	20 m/48 h
[32]		POST	NO	IMPAIRMENT	6 h
[34]		POST	NO	NO EFFECT	24 h

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
