# Peer review of "A Systematic Review of the Dietary Choline Impact on Cognition from a Psychobiological Approach: Insights from Animal Studies"

_nutrients, 2021, doi:10.3390/nu13061966_

Round 1

Reviewer 1 Report

The Authors Gamiz and Gallo submitted a summary-like review describing the research performed during the last two decades on the relevance of dietary choline in cognition.

Despite being descriptive and summary-like review, it could offer a valuable and clear reference for the researcher working in the same field.

Author Response

We thank the reviewer for the positive feedback. Please see the attachment with the complete response letter to the reviewers. 

Reviewer 2 Report

This is a very interesting paper. The authors do an excellent job of summarizing the literature within the bounds of laid out in the aims.

I think whats missing is some more analysis that ties the results from the literature together.

For example:

In the abstract, the authors describe dietary choline as being relevant to cognition, but perhaps a sentence or two saying how it is relevant is merited.

In the introduction, the authors describe other reviews done on choline and cognition, but on other sets of studies.

The authors should describe clearly the conclusions of those reviews and state clearly why it is important to conduct a review of mouse studies.

It doesnt sound right just to state that it should be done simply because it hasnt been done before. What does do the authors expect to find by conducting such a review, that wasnt known before?

Likewise at the conclusion, some call to action needs to be articulated, therefore I believe 3 pieces are missing:

1) What tentative model regarding the relationship between choline and cognition can be drawn (linking as much as possible to the physiological, form biochemical pathways, to brain function, to effects on congnition). A diagram might be useful to show this model and describe which parts are more certain and which are uncertain.

And some description of what kinds of studies would be necessary to remove the uncertainties.

The authors do state that more research should be done, etc. But what should that research be? Even tentative hypotheses would be helpful to draw together the material.

In line 828, the authors mention "critical issues" to be considered in human studies. Could the authors explain a little further how the animal models should impact future human studies?

Author Response

Dear Reviewer 2,

we want to acknowledge your helpful suggestions. 

Please see the attachment with the response to reviewer comments. We hope that all the points have been addressed and reviewer’s text had not been unintentionally cut off by the system since the reviewer mentions 3 missing points and lists only one. We look forward to your final decision regarding our revised submittal in due time.

Sincerely,

Fernando Gámiz

Round 2

Reviewer 2 Report

The authors answered all the reviewers' concerns well. The well-designed diagram, in particular, brings the model described to life. Also the review now describes future studies that should be carried out.

I recommend accepting this manuscript for publication.